# Flexible Abrasive Tools for the Deburring and Finishing of Holes in Superalloys

**Adrián Rodríguez, Asier Fernández, Luís Norberto López de Lacalle**  **and Leonardo Sastoque Pinilla \***

Aeronautics Advanced Manufacturing Center, 48170 Zamudio, Spain; Adrian.Rodriguez@ehu.eus (A.R.); Asier.Fernandez@ehu.eus (A.F.); Norberto.lzlacalle@ehu.eus (L.N.L.d.L.)

**\*** Correspondence: cfaa2015@ehu.eus or EdwarLeonardo.Sastoque@ehu.eus; Tel.: +34-688-673-836

**Abstract:** Many manufacturing sectors require high surface finishing. After machining operations such as milling or drilling, undesirable burrs or insufficient edge finishing may be generated. For decades, many finishing processes have been on a handmade basis; this fact is accentuated when dealing with complex geometries especially for high value-added parts. In recent years, there has been a tendency towards trying to automate these kinds of processes as far as possible, with repeatability and time/money savings being the main purposes. Based on this idea, the aim of this work was to check new tools and strategies for finishing aeronautical parts, especially critical engine parts made from Inconel 718, a very ductile nickel alloy. Automating the edge finishing of chamfered holes is a complicated but very important goal. In this paper, flexible abrasive tools were used for this purpose. A complete study of different abrasive possibilities was carried out, mainly focusing on roughness analysis and the final edge results obtained.

**Keywords:** flexible abrasive tools; finishing; rounding edge; superalloys

## 1. Introduction

Titanium alloys and nickel-based superalloys are widely used today in aerospace components, commonly used in engines, considering that superalloys and concretely Inconel 718 are capable of working in corrosive environments and at high temperatures. Those materials can be used as part of gas turbine engines, steam, nuclear components, chemicals, etc. There is a strong demand for dimensional accuracy and surface roughness for these high-value components.

Drilling holes in aerospace components is often a delicate operation; the hole amplifies the stress around it by a factor of two [1]. Moreover, it is often the last machining operation, with a looming risk of making a scrap part due to a single bad hole. This circumstance determines the final time used in the production of the part, and a lack of quality can lead to its rejection, as it should especially take into account the reliability of the process due to the costs already involved. Therefore, it is a high value-added operation [2].

Currently in industrial practice, drilling processes are widely used due to their versatility and the short time invested in performing the task. However, these operations produce results of not very high quality, thus requiring complementary operations such as dotting, re-drilling, reaming, chamfering and edge finishing. This fact supposes a waste of time, both in subsequent cutting processes and in tool changes. The "not very high quality" refers, basically, to the deviations that occur in terms of diameter tolerances, surface roughness and burr formation, which are inherent phenomena in the process. Also, the effect of the cutting parameters on the hole quality (circularity and hole diameter) and tool wear during the drilling of super alloy Inconel 718 allows us infer that the cutting speed and feed rate played a large role in the variation of the deviation from the circularity values [3]. The available literature

regarding drilling in high strength materials is rather limited [4,5]. However, in recent years there have been further investigations into new techniques and processes to drill holes in these alloys.

Among these new techniques, ultrasonic assisted machining is one of the most commonly used. This is a machining technology where a high frequency vibration (20 kHz) with an amplitude around 10 μm overlaps the continuous movement of the cutting tool, providing an output power between 50 W and 3000 W [6]. The use of ultrasonic-assisted processes allows a reduction in cutting forces by 30–50% [7], an improvement of the final surface quality, better chip evacuation and a longer tool life [8].

Other authors propose alternatives to traditional drilling. The idea is to use a ball-end milling tool giving it a helical motion around the hole. Regarding the helical milling, there are two similar helical milling techniques: Ball helical milling (BHM) and contouring ball helical milling (CBHM) [9]; the results were quite good in terms of quality but the times were far from those obtained with twist-drilling operations, or in other processes [10,11]. Takt-time in aeroengine manufacturing in many cases prevents the replacement of drilling with twist drills, thus edge burrs and poor finishing are common issues. In emerging processes, the plasticity of metal is also a key factor, as shown in [12,13].

In this paper, brushing techniques using abrasive flexible tools are studied. The aim is to implement these tools for the finishing process, to improve the surface finish obtained on the one hand, and to achieve the rounding of the edges in the countersunk holes on the other. Flexible hone tools are available in silicon carbide, aluminum oxide, zirconia alumina, boron carbide, tungsten carbide and even in other grades, with diameters ranging from 4 to 1000 mm.

In this work, different available state-of-the-art tools are presented. Tests were carried out in order to make a first attempt to use these tools, with interesting results that are shown below.

## 2. Flexible Abrasive Tools

Companies such as Brush Research Manufacturing (BRM) have a long history of solving difficult finishing problems with brushing technology. The term "brush" is commonly associated with classic twisted wire brushes or the nylon brushes used for deburring or edge blending. It is a flexible and elastic abrasive tool, ideal for soft cutting in finishing operations, "plateau honing", cylinder liner deburring, hydraulic and pneumatic components, as well as other industry sectors such as aeronautics, automotive parts, screw machining, etc.

These are a general-purpose tools (Figure 1), the versatility of which stems from the small abrasive balls overlapping at the end of a nylon filament. Each ball is independent of the others; this fact ensures the centering and auto-alignment with the hole. Having complete control of the process parameters and identifying and assessing the influence on the final surface is essential for the efficient implementation of these tools in CNC machines and robots.

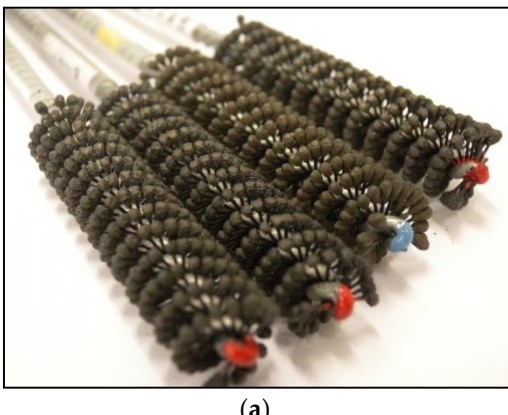 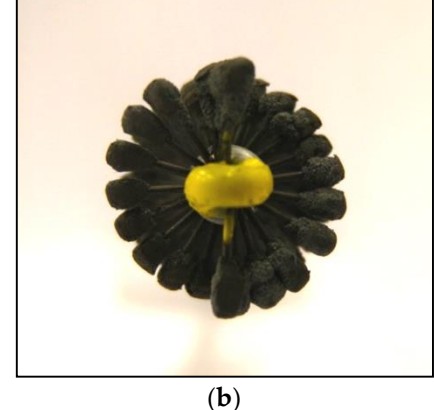

(**a**)        (**b**)

**Figure 1.** (**a**) Tools with different abrasive qualities. (**b**) Tool detail (3×).

One application of these flexible tools is the surface finishing and edge blending of holes made in aeronautic alloys, such as Inconel 718 and Ti6Al4V. A wide range of abrasives and grit sizes are offered by BRM and other companies. This implies the necessity to carry out a comparison between the different abrasive grades. The test results for different abrasive types and grain sizes are presented in this work. The parameter measured in this first approach was the final roughness of the brushed holes. Table 1 shows the variety of tools used. (Prices are shown because in some cases these are twice those of other solutions.)

**Table 1.** Different flexible abrasive tools used in tests.

| Code | Abrasive | Grit Sizes | Nominal Diameter | Web Price |
|---|---|---|---|---|
| SC10 | Silicon Carbide | 180 | 10 mm | ≈12 $/un |
| SC11 | Silicon Carbide | 180 | 11 mm | ≈12 $/un |
| SC11-400 | Silicon Carbide | 400 | 11 mm | ≈23 $/un |
| BC11 | Boron Carbide | 180 | 11 mm | ≈15 $/un |
| Di11 | Diamond | 2500 C Mesh | 11 mm | ≈30 $/un |

## 3. Previous Tests on Ti6Al4V Alloy

Preliminary tests on Ti6Al4V alloy were carried out. This alloy was used firstly because is more economic, easier to buy and has better machinability than the superalloys, such as Inconel 718. Titanium plates were used (200 × 100 × 7.5 mm dimensions); the main aim of these tests was to establish a first approach to the process before carrying it out on Inconel 718. On the other hand, titanium alloys are used not only in engines but in airframe key parts, which join the wings to the airplane body.

Prior to conducting the brushing tests, 80 holes of 10.7 mm diameter were drilled in the plates. The following conditions were used for the drilling: Vc = 35 m/min and f = 0.12 mm/rev. Figure 2 shows the experimental set-up.

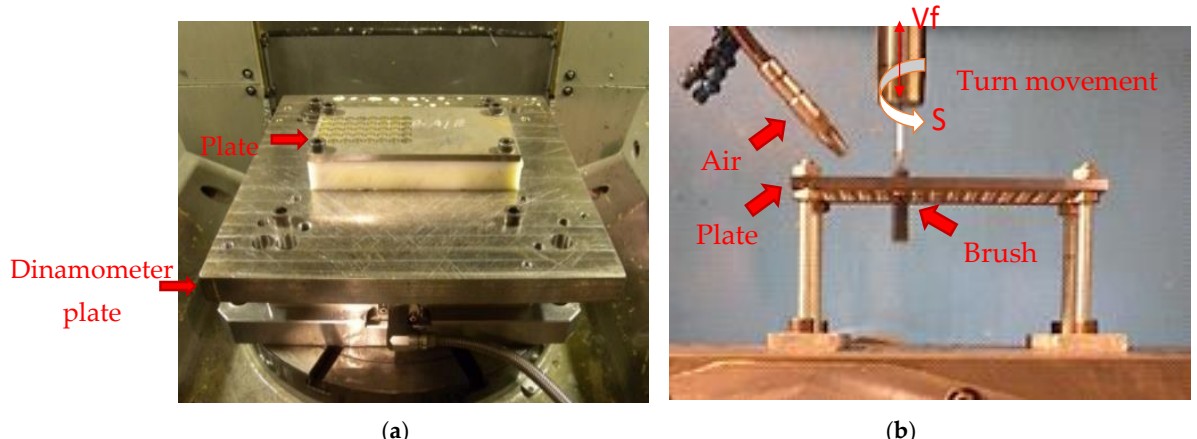

(**a**) (**b**)

**Figure 2.** (**a**) Drilling set-up. (**b**) Brushing set-up.

As shown in Figure 1 and Table 1, five different flexible abrasive tools were used. With each different brush, 16 holes were made at these brushing conditions: Vc = 60 m/min and f = 0.5 mm/rev. Figure 3 shows the results of the roughness measurement, both for the drilled holes and the brushed holes. Firstly, the surface quality obtained in the previous drilling with the conditions used was quite good, averaging around 0.5 μm Ra. The main problem was the results dispersion, with varying Ra roughness parameters from 0.29 to 1.17 μm.

After brushing with the five different flexible abrasive tools, the roughness parameters decreased and the results dispersion was lower. The best brush type for this material in terms of surface roughness

was the SC11-400, as it reduced the average Ra roughness up to 0.25 µm, with values between 0.2 µm and 0.3 µm (Figure 3).

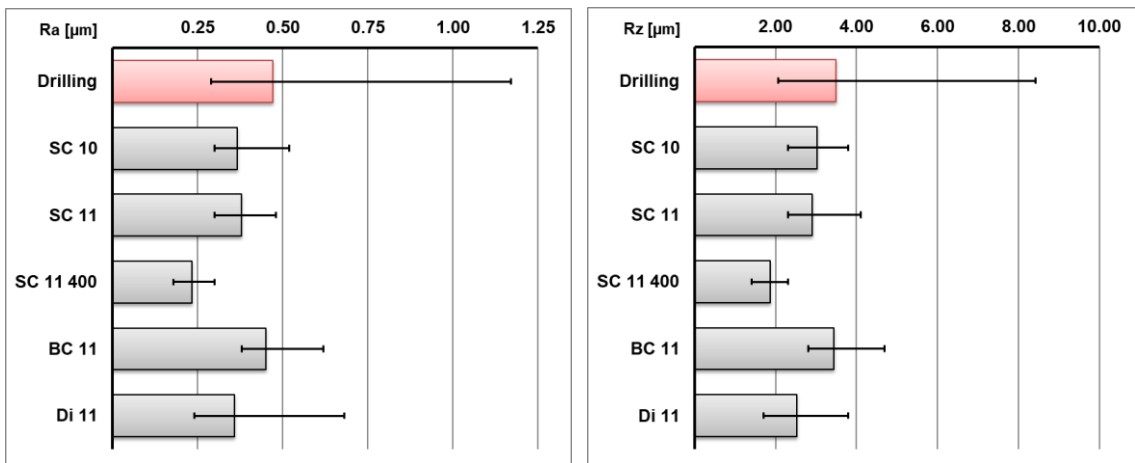

**Figure 3.** Ra [µm] and Rz [µm] roughness values after drilling and after finishing Ti6Al4V.

## 4. Test on Inconel 718 Superalloy

Based on data from the preliminary tests carried out on Ti6Al4V, we found that that the five different flexible abrasive brushes were able to reduce the roughness parameters. Moreover, it was an easy and economical finishing process that can be carried out using machine tools.

On the other hand, the preliminary tests showed a low cutting capacity. It was difficult to make a chamfer on a hole or deal with large burrs because removing that much material was impossible. However, these brushes could be useful in order to finish surfaces, round edges or carry out cross-hole deburring [9]. For these reasons, experimental tests were carried out on Inconel 718 plates, which is a commonly-used material in aerospace components working at high temperatures. This is a difficult material to machine, so the soft cut of these brushes may have been insufficient.

In this case, Inconel 718 plates were used with the dimensions $200 \times 100 \times 7.5$ mm, similar to those used in the preliminary tests in titanium. The tests were carried out in an Ibarmia ZV25 milling machine, with a spindle with 25 KWs. Regarding the previous drilling, the Table 2 shows the two different conditions used.

**Table 2.** Conditions used for drilling in Inconel 718.

|  | Vc [m/min] | f [mm/rev] | S [rpm] | Vf [mm/min] | No. Holes |
|---|---|---|---|---|---|
| "A" Conditions | 20 | 0.06 | 595 | 35.7 | 40 |
| "B" Conditions | 25 | 0.06 | 744 | 44.6 | 40 |

### 4.1. "A" Conditions

Two different cutting conditions for previously-drilled Inconel 718 were tested. The first conditions were established by the tool manufacturer, the second ones were rather demanding in order to reduce the processing time and increase productivity. The aim was to compare the surface roughness results after brushing. The drilling parameters in the "A" conditions were c = 20 m/min and f = 0.06 mm/rev.

After performing the drilling, brushing tests were carried out. In this case the same brushing conditions as in the preliminary test were used (Vc = 60 m/min and f = 0.5 mm/rev).

Figure 4 shows the roughness results obtained before and after brushing. The surface roughness obtained after drilling was moderately good (around 0.5 µm Ra) thanks to the conservative drilling conditions used. After brushing, the results showed that the roughness values decreased somewhat, but not significantly. In the case of the SC10 brush, the roughness became worse. This implies that for

this material and using the drilling conditions given by the manufacturer, these SC10 brushes were not suitable. As for the rest of the tests, similar to the results for the titanium, the best roughness results were achieved with SC11-400 brushes. However, the BC11 brush provided similar roughness values and less deviation in the results. In addition, the BC11 brushes were cheaper and showed less wear after brushing than SC11-400, thus BC11 was the most suitable in this case.

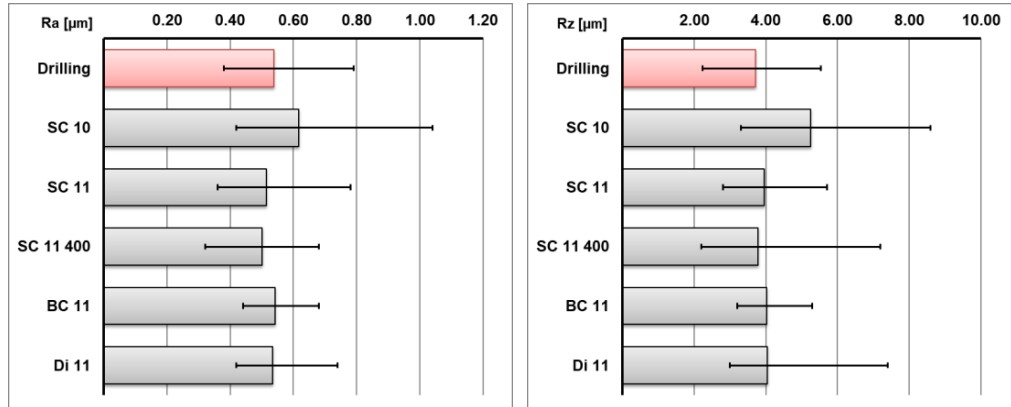

**Figure 4.** Ra [μm] and Rz [μm] roughness values after drilling and finishing Inconel 718.

The results obtained in this case also show that under these drilling conditions it was too unproductive to execute a brushing operation because the surface improvement was hardly noticeable. In the following section, the brushing process in more demanding drilling conditions is examined. In this case, the brushing process might be useful.

*4.2. "B" Conditions*

In this section, the previous drilling conditions were Vc = 25 m/min and f = 0.06 mm/rev. In this way, the roughness results obtained after drilling were worse than in the previous cases. However, brushing could be useful in this case. Figure 5 shows the roughness results. The roughness values observed before brushing were around Ra 0.9 μm, with a large dispersion of results. After brushing, the roughness parameters decreased to values lower than 0.65 μm Ra. In this case, the BC11 brush provided the lowest roughness values and the lowest deviation of the results, thus this was the most convenient brush. Besides, the tool wear was not critical in this case.

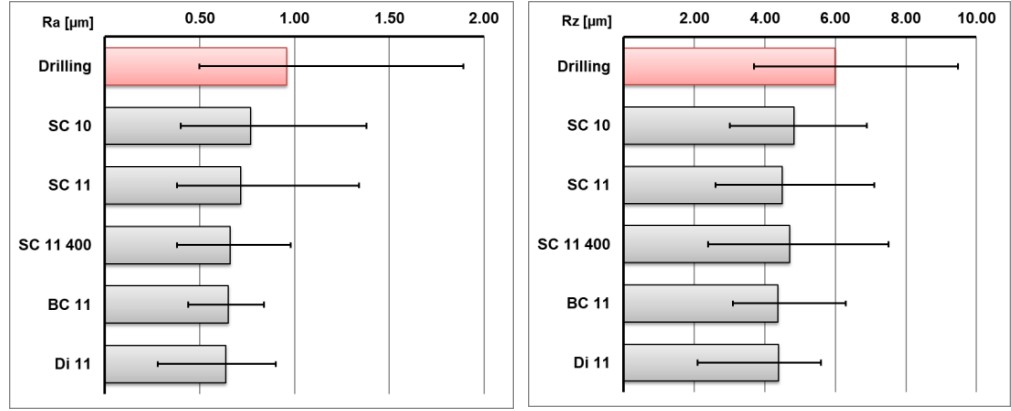

**Figure 5.** Ra [μm] and Rz [μm] roughness values after drilling and after finishing Inconel 718.

It is noted that the results showed that the cutting ability of these brushes was limited, especially when cutting materials with low machinability such as Inconel 718. Therefore, a large amount of deburring and chamfering of holes was impossible. However, once the hole was chamfered, rounding

the edges and finishing the surface was achieved using the flexible abrasive brushes [11]. Figure 6 shows one of the drilled and brushed holes. The process involved drilling, chamfering and brushing. Figure 6 and Table 3 set out the rounding edge produced by the brushes.

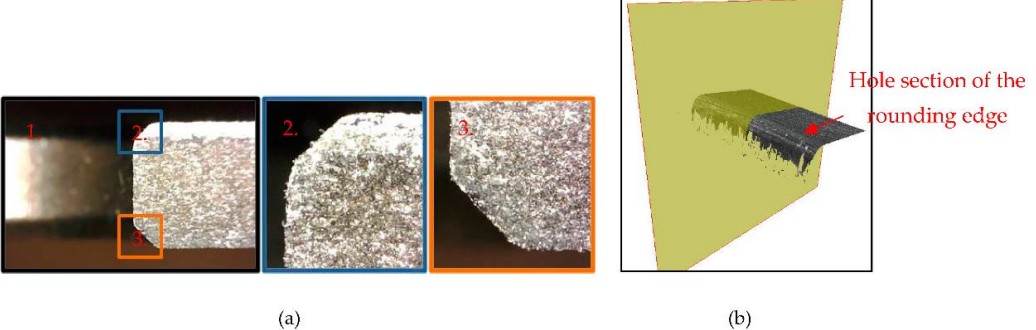

| (a) | (b) |

**Figure 6.** (**a**) Hole section and detail of the chamfer and rounding edge. Image 1 (5×), Image 2 (10×), Image 3 (10×). (**b**) Angle detail measured with optical means of the rounding edge of a hole section (5×).

**Table 3.** Angle detail—hole section of the rounding edge.

|  | Angle/° | Apex X/mm | Apex Y/mm |
|---|---|---|---|
| Angle 1 | 102.241 | 99.452 | 48.341 |

For years, the edge finishing process has been by hand in many areas, but now the tendency is to try to automate these finishing processes [12]. One automation possibility uses flexible abrasive brushes. However, others are possible, such as using shape tools. In this case, the main drawback is the correct tool positioning and also the tangents to the surface. In addition, it is necessary to consider the fact that many of these holes are placed in curved areas or in areas that are difficult to access for a conventional milling tool. The main problem with the brushes is the lack of repeatability and the rapid wear suffered.

Figure 7 shows some photographs of the brushes following their use. As mentioned above, despite achieving the best results, the SC11-400 brush is one of the most expensive, along with the diamond brush. Besides, tool wear on these brushes is greater than the other brushes. To conclude, regarding tool wear, BC11 is the most appropriate option for materials such as Inconel 718. Moreover, in some cases it is also the best option in terms of the surface quality achieved.

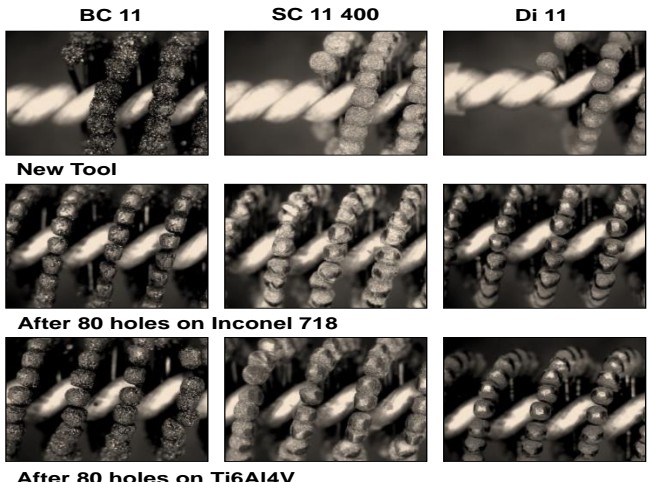

**Figure 7.** Tool wear on three different brushes. New tool; after Inconel 718; after Ti6Al4V (3×).

## 5. Automatic Process

Polishing and deburring is a process with great automation possibilities, through the use of robots [13]. Force feedback control is the key aspect to be considered. The definition of a robotic cell for the application of the process is based on accessibility. The flexible brushes being applied to holes in different aeroengine components and cases being produced with Inconel 718, Hastelloy or other nickel alloys are a good task for these robots. Deburring and edge finishing will be always a step in the process chain. The idea proposed is to work in high-automation mode, in the following stages:

- Burr detection, for instance using structured blue light or other optical means. The random location implies a random pattern.
- Robotic deburring in brush manipulation: a robotic arm can use a spindle with the usual low-torque to brush at the required rotational speed.
- Final check: optical means will help, and in cases of internal hole surface roughness, a roughness meter measurement is necessary.

The proposed system for the automation process consists of a unique superfinished cell capable of working in two different work modes, in particular with a tool on a robot or with a piece on a robot. In the first of the work modes (MOD1), the idea was to work on pieces of large dimensions (Ø 2400 mm, height 1500 mm, weight 2500 kg) mounted on a rotating table and working with tools mounted on the robot, which is able to access the outer and inner areas of the type pieces. The materials to be worked in this case will be heat-resistant alloys with mechanical characteristics equal to or higher than Inconel 718, Titanium 6-4, Jethete type stainless steel or similar. The operations to be carried out will be diverse, highlighting operations of deburring, edge killing and polishing of localized areas and holes, as well as measurement and control operations. In a second mode of work (MOD2), we will work with tools mounted in fixed posts, with the piece positioned mounted on the manipulator robot. In this case, the pieces to be treated will be units or sets coming from castings or other types of components with the maximum dimensions of approximately 1000 × 1000 × 1000 mm and maximum weights of up to 120 kg. The materials will have characteristics similar to those indicated for the MOD1 work mode and the operations to be carried out include cutting and sanding processes as well as other operations such as those already mentioned for deburring, polishing and measuring and control. Figure 8 shows the system used to apply the approach.

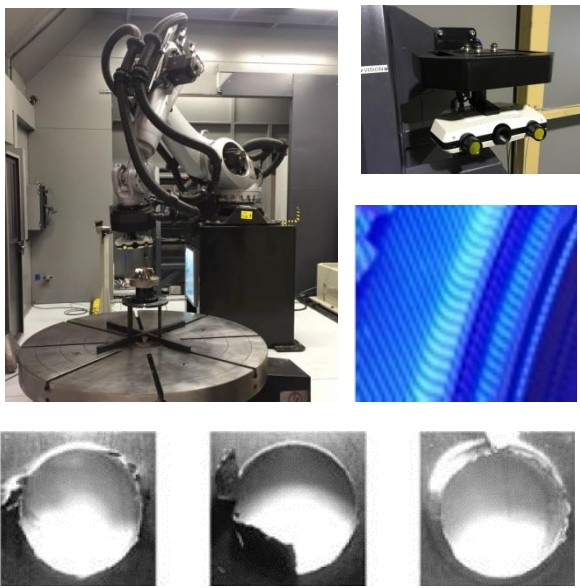

**Figure 8.** Robotic deburring: robot arm, structured-blue light devices, detail of holes with burrs.

## 6. Conclusions

Several contributions of this research can be pointed out, namely:

- Preliminary tests on Ti6Al4V show that flexible abrasive brushes are able to reduce the roughness parameters of drilled holes. Furthermore, the final roughness shows less deviation from the average value in comparison to the previously-drilled holes. In this material, considering the roughness values, Silicon Carbide 400 grit size brushes were the most suitable.
- Despite the fact that the brushes were not suitable for chamfering or removing large burrs, tests made on Inconel 718 showed that these brushes could be a great option for rounding edges and surface finishing. Particularly, BC11 brushes were the most suitable for this operation. After brushing with BC11, the roughness was better, the deviation of results was lower, and their price and wear resistance make them suitable for this aim.
- Brushes are a real choice in the industrial environment in order to achieve a rapid and efficient improvement of the inner quality of holes and to eliminate burrs at the hole edge, both at the entrance and exit of the drill bit from plates.
- Polishing, deburring, burr detection or a final check by optical means for large pieces are processes with great automation possibilities by robotic means.

**Author Contributions:** Conceptualization, A.R., A.F. and L.N.L.d.L.; Methodology, A.R.; Software, A.R. and A.F.; Validation, A.R. and A.F.; Formal Analysis, A.R. and L.N.L.d.L.; Investigation, A.R. and A.F.; Resources, L.N.L.d.L.; Data Curation, L.S.P.; Writing-Original Draft Preparation, A.R.; Writing-Review & Editing, L.S.P.; Visualization, L.S.P.; Supervision, A.R. and L.N.L.d.L.; Project Administration, L.N.L.d.L.; Funding Acquisition, L.N.L.d.L.

**Funding:** The authors gratefully acknowledge the project "Estrategias avanzadas de definición de fresado en piezas rotativas integrales, con aseguramiento de requisito de fiabilidad y productividad IBRELIABLE" (DPI2016-74845-R), and "Discos de freno premium para trenes de alta velocidad", by the Spanish Ministry of Economy.

**Conflicts of Interest:** The authors declare no conflict of interest.

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
