# Peer review of "Flexible Abrasive Tools for the Deburring and Finishing of Holes in Superalloys"

_jmmp, doi:10.3390/jmmp2040082_

Reviewer 1 Report

1.       Please add one figure to shown the movement during this polishing process, combining with the movement parameter, such as feeding rate Vf, etc.

2.       What kind of device is used to measure the roughness, and where are the measurement positions.

3.       Please add scale bar in Fig. 6. The description of fig. 6(b) is not clear, pls add more information in the figure.

4.       Check the reference carefully. Where is Ref. 12.

Author Response

Good morning,

First of all, thank you for your kind and conciousness revision of our article. 

Second, let us respond to your report point by point:

The figure 2 was already actualized, more information was given to the both images in the figure.

The measurements were made with a portable surface roughness tester Mitutoyo Surftest SJ-210 (the specifications are shown in the Table 1 of the attached document).  The plotted Roughness value is the average of the values of the measurements of all the holes made with each brush / parameter. Each hole was measured in two positions, that is, at 0º and 180º.

However, we also consider that this information about the configuration and device used to measure the roughness its not that relevant to the content and regular understanding in the article and we will not included there.

The Figure 6 was already actualized following your kind report.

This was an honest mistake which its already corrected.

Also, the conclusions was revised and actualized, and some more regarding the automation process and the hole drilling process was included.

Reviewer 2 Report

Review of 390424-peer-review-v1: “Flexible abrasive tools for deburring and finishing of holes in superalloys”

In this paper, brushing techniques using abrasive flexible tools are studied. Tests were carried out in order to make a first approach to the use of these tools.

The subject of the paper is relevant with the topics of the journal and this topic has a great deal of value for industry, especially for aeronautical applications and use.

The references are well selected and used. I would expect some more to be used because the area of quality in drilling holes is quite extensive. I would suggest the authors to separate the introduction paragraph with the literature survey (state of the art) paragraph, and then increase the number of the references used.

The number of the cutting condition used is limited for manufacturing processes studies. Nevertheless, as the authors stated, this paper is aiming in starting the ball rolling (“…to make a first approach to the use of these tools”).

The paragraph “5. Automatic process” should be extended, commenting more in the application of the proposed process within a robotic cell.

My proposal to the editor is to accept with minor revisions.

Author Response

Good morning and thank you so much for your kind words and consciousness report. Up next, we will respond to your comments and suggestions.

Regarding to the references about articles in the subject, two more were include in the introduction of the Article, and the information reorganized for a better presentation and understanding.

The information about the automatic process was extended, including the main characteristics of the robotic cell that was mentioned in that very same chapter.

Once again, thank you so much for your words, and we hope that this changes made in the article would be suitable for you.

Have a nice day.

Round  2

Reviewer 1 Report

I accept the revision.